# Comparative analysis of transcriptomic profiles among ascidians, zebrafish, and mice: Insights from tissue-specific gene expression

**Shin Matsubara***, **Tomohiro Osugi**, **Akira Shiraishi**, **Azumi Wada**, **Honoo Satake**

Bioorganic Research Institute, Suntory Foundation for Life Sciences, Kyoto, Japan

* matsubara@sunbor.or.jp

**Data Availability Statement:** All relevant data are within the manuscript and its Supporting Information files.

## Abstract

Tissue/organ-specific genes (TSGs) are important not only for understanding organ development and function, but also for investigating the evolutionary lineages of organs in animals. Here, we investigate the TSGs of 9 adult tissues of an ascidian, *Ciona intestinalis* Type A (*Ciona robusta*), which lies in the important position of being the sister group of vertebrates. RNA-seq and qRT-PCR identified the *Ciona* TSGs in each tissue, and BLAST searches identified their homologs in zebrafish and mice. Tissue distributions of the vertebrate homologs were analyzed and clustered using public RNA-seq data for 12 zebrafish and 30 mouse tissues. Among the vertebrate homologs of the *Ciona* TSGs in the neural complex, 48% and 63% showed high expression in the zebrafish and mouse brain, respectively, suggesting that the central nervous system is evolutionarily conserved in chordates. In contrast, vertebrate homologs of *Ciona* TSGs in the ovary, pharynx, and intestine were not consistently highly expressed in the corresponding tissues of vertebrates, suggesting that these organs have evolved in *Ciona*-specific lineages. Intriguingly, more TSG homologs of the *Ciona* stomach were highly expressed in the vertebrate liver (17–29%) and intestine (22–33%) than in the mouse stomach (5%). Expression profiles for these genes suggest that the biological roles of the *Ciona* stomach are distinct from those of their vertebrate counterparts. Collectively, *Ciona* tissues were categorized into 3 groups: i) high similarity to the corresponding vertebrate tissues (neural complex and heart), ii) low similarity to the corresponding vertebrate tissues (ovary, pharynx, and intestine), and iii) low similarity to the corresponding vertebrate tissues, but high similarity to other vertebrate tissues (stomach, endostyle, and siphons). The present study provides transcriptomic catalogs of adult ascidian tissues and significant insights into the evolutionary lineages of the brain, heart, and digestive tract of chordates.

## Introduction

During the past two decades, genome assembly and phylogenetic analyses of *Ciona intestinalis* Type A (or *Ciona robusta*) have verified that ascidians belong to the Urochordata phylum,

**Funding:** This work was supported by the Japan Society for the promotion of Science (http://www.jsps.go.jp/english/index.html) to Shin Matsubara (JP19K16182). The funder had no roles in study design, data collection and analysis, decision to publish, or preparation of the manuscript.

**Competing interests:** The authors have declared that no competing interests exist.

which are the closest living relatives to the Vertebrata phylum in the Chordata superphylum [1–3]. Due to their important phylogenetic position, ascidians have attracted attention as model organisms for evolutionary studies. Particularly, the simplicity of the larval body and experimental advantages of *in vitro* fertilization and embryogenesis have revealed various conserved features in the development of the central nervous system (CNS) [4–8]. Moreover, recent single-cell analyses further detailed the cell fates reported in previous studies and revealed conserved gene regulatory networks during embryo development [7–9]. Such studies have underscored the morphological and developmental similarities between ascidian larva and vertebrates and provided significant insights into the evolutionary lineages of embryogenesis and morphogenesis in chordates [4–6].

Adult ascidians (or sea squirts) develop from swimming larvae via metamorphosis and 1st and 2nd ascidian stages in 2.5–3 months [10]. They are enveloped by a polysaccharide-containing tunic and intake water and food from an oral siphon (Fig 1) [11]. The pharynx serves dual roles as an apparatus for food collection and gas exchange with water (Fig 1) [11]. The endostyle is located at the ventral side and secretes mucus into the pharynx. The resultant food cord is transported to the stomach and intestine, and then excreted through the atrial siphon (Fig 1) [11]. Mature oocytes are produced in the ovary located beside the heart and are spawned from the atrial siphon via an oviduct (Fig 1) [11]. These peripheral tissues are regulated by the neural complex (Fig 1) [11]. In fact, we have identified more than 30 neuropeptides and visualized the entire neural network of adult ascidians using transgenic animal models [12–15]. In contrast, despite the growing body of knowledge regarding early embryos and larvae, less attention has been paid to the functions and evolutionary lineages of the adult tissues of ascidians.

Cell types and functions in an organism are featured (or characterized) by specific gene expression. Thus, tissue-specific genes (TSGs) are important for tissue-specific function and/or development. In vertebrates, expression profiles for various TSGs are more conserved in the same or functionally related organs among different species than in other organs in the same species [16–19]. Moreover, the speed of evolution is fast in paralogs but slow in orthologs [20, 21]. Therefore, identification of TSGs in *Ciona* and comparison to their vertebrate homologs is expected to significantly contribute to the clarification of the evolutionary origin and functional lineages of the respective tissues in chordates. In *Ciona*, expressed sequence tag data during embryogenesis [22, 23] and in young adults [24], and transcriptomes of young and adult ovaries and isolated ovarian follicles [25, 26], are available. Moreover, microarray data for adult tissues have identified TSGs in 11 tissues and determined their chromosomal locations [27]. However, the lack of comparative analyses of *Ciona* TSGs with vertebrate homologs has hindered the understanding of the evolutionary implication in each tissue in chordates.

In this study, we present transcriptomic profiles for adult tissues of *Ciona intestinalis* Type A (or *Ciona robusta*) and identify the TSGs in each tissue. Searching for zebrafish and mouse homologs of the *Ciona* TSGs and analyzing their tissue distributions in zebrafish and mice uncovered gene expression similarities in each tissue among the species. Such comparative analyses of *Ciona* TSGs with their homologs in zebrafish and mice provide evolutionary insights into the biological functions of *Ciona* tissues.

## Results and discussion

### Identification of TSGs in *Ciona*

To identify TSGs in *Ciona*, RNA-sequencing (RNA-seq) was performed using 11 samples of 9 tissues of adult ascidians (oral siphon, atrial siphon, neural complex, endostyle, heart, ovary, pharynx, stomach, and intestine). The raw sequence data were deposited into the NCBI database (PRJNA731286). Total reads, mapping rates, and accession numbers for each sample are

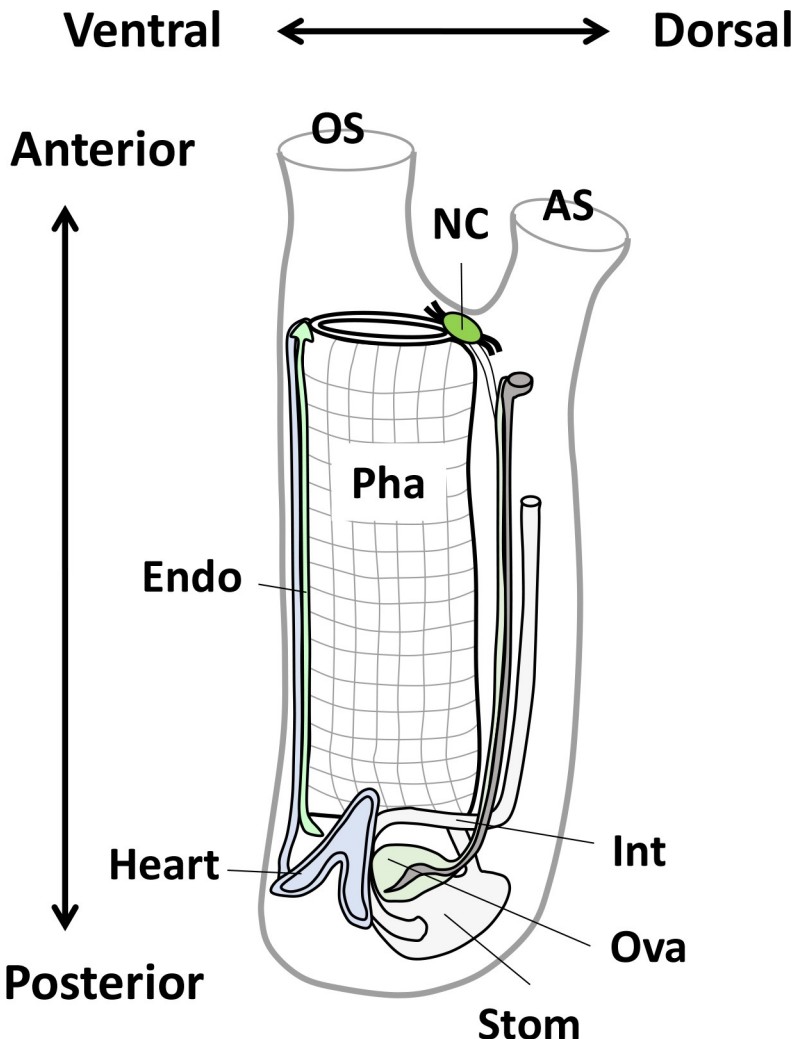

**Fig 1. Schematic illustration of the body structure of an adult ascidian.** The illustration was modified from Osugi et al., 2020 [12]. The key anatomical parts of the 9 tissues analyzed in this study are indicated. AS, atrial siphon; Endo, endostyle; Int, intestine; NC, neural complex; OS, oral siphon; Ova, ovary; Pha, pharynx; Stom, stomach.

listed in Table 1. The expression level for each gene was calculated as RPKM (reads per kilobase per million total reads) and is listed in S1 Table with raw read numbers. A previous microarray analysis identified TSGs based on the ratio of the median expression values between a specific organ and others (cutoff values ranging 1.3–9.4) [27]. Such median-based identification includes specifically expressed genes in individual and multiple tissues (e.g., brain- and intestine-specific genes). To define tissue specificity more strictly, we identified TSGs based on the RPKM (RPKM > 1 in an individual tissue and RPKM < 0.5 in all other tissues, i.e., all TSGs show more than 2-fold higher expression than any other tissues) (Tables 1 and S2). The RNA-seq data reproduced some of the specific expression patterns of the previously identified TSGs in the neural complex, endostyle, heart, ovary, stomach, and intestine (S1 Fig) [27], confirming the reliability of the RNA-seq data and their usefulness for the following analyses. The greatest number of TSGs was identified in the intestine (312 genes) and the least was in the pharynx (15 genes) (Table 1). Amino acid sequences for the identified TSGs in *Ciona* were subjected to BLASTP analysis against the RefSeq protein database for mice and

**Table 1. RNA-seq summary of adult *Ciona* tissues.**

| Sample | Number of TSG | Total reads | % Mapped | Accession |
| --- | --- | --- | --- | --- |
| Oral Siphon | 112 | 28,379,690 | 94.92 | SRR14597452 |
| Atrial Siphon | | 22,999,973 | 94.92 | SRR14597461 |
| Neural Complex | 97 | 23,236,898 | 89.91 | SRR14597453 |
| Endostyle | 66 | 25,164,333 | 92.97 | SRR14597460 |
| Heart | 50 | 22,512,838 | 91.20 | SRR14597459 |
| Ovary | 180 | 23,763,137 | 94.48 | SRR14597457 |
| Pharynx | 15 | 23,396,141 | 87.68 | SRR14597458 |
| Stomach | 38 | 28,571,477 | 91.70 | SRR14597456 |
| Intestine (proximal) | 312 | 24,346,598 | 91.20 | SRR14597451 |
| Intestine (middle) | | 26,999,786 | 89.67 | SRR14597454 |
| Intestine (distal) | | 26,160,308 | 89.46 | SRR14597455 |

TSGs for the siphons and intestine include the genes with RPKM > 1 either in the oral siphon or atrial siphon and in any part of the intestine, respectively. The RNA-seq reads were mapped to the *Ciona* cDNA library, and the mapping rates and accession numbers were shown.

zebrafish using an e-value threshold < 1e-5. More than 50% of the TSGs in the siphons (66 genes, 58.9%), neural complex (56 genes, 57.7%), heart (31 genes, 62.0%), ovary (105 genes, 58.3%), and stomach (23 genes, 60.5%) were found to be homologous to the mouse and/or zebrafish genes (Fig 2). On the other hand, 37 genes (56.1%) in the endostyle, 204 genes (65.4%) in the intestine, and 12 genes (80.0%) in the pharynx had no homologous mouse or zebrafish genes, suggesting that these tissues have *Ciona*-specific functions (Fig 2).

## Similarity of gene expression patterns between *Ciona* TSGs and their homologs in mice or zebrafish

The tissue distributions of mouse and zebrafish genes homologous to the *Ciona* TSGs were investigated using RNA-seq data of 17 mouse organs (30 tissues, PRJNA267840) and 12 zebra-fish tissues (PRJNA255848). RPKM-based tissue specificity of the *Ciona*-TSG homologs in the vertebrates was also investigated in the same criteria as *Ciona*. Although 13 vertebrate homologs of the *Ciona* TSGs in the neural complex were also tissue-specific in the vertebrate brain, none or a small number of vertebrate TSGs shared tissue specificity with *Ciona* TSGs in the other tissues (Fig 2 and S2 Table). This indicates that tissue specificity of the gene expression is not necessarily conserved among *Ciona*, zebrafish, and mice except for the neural complex and brain. The expression levels of vertebrate homologs of the *Ciona* TSGs were then normalized from 0 to 1 and the number of highly expressed genes (> 0.8) in each tissue was counted. The ratio was indicated as "tissue similarity" between *Ciona* and zebrafish or mice. Based on the tissue similarity, *Ciona* tissues were categorized into the following three groups. (i) TSG-rich tissues homologous to vertebrate counterparts: high similarity to the corresponding tissues in mice and zebrafish, (ii) *Ciona*-unique gene-rich tissues: low similarity to the corresponding and/or other tissues, and (iii) homologous TSG-rich tissues histologically unrelated to the vertebrate counterparts: low similarity to the corresponding tissues but high similarity to other tissues.

## (i) TSG-rich tissues homologous to vertebrate counterparts: Neural complex and heart

The mouse (62.5%) and zebrafish (48.2%) homologs of *Ciona* neural complex-specific genes showed high expression in the corresponding mouse and zebrafish brains (Fig 3A), indicating

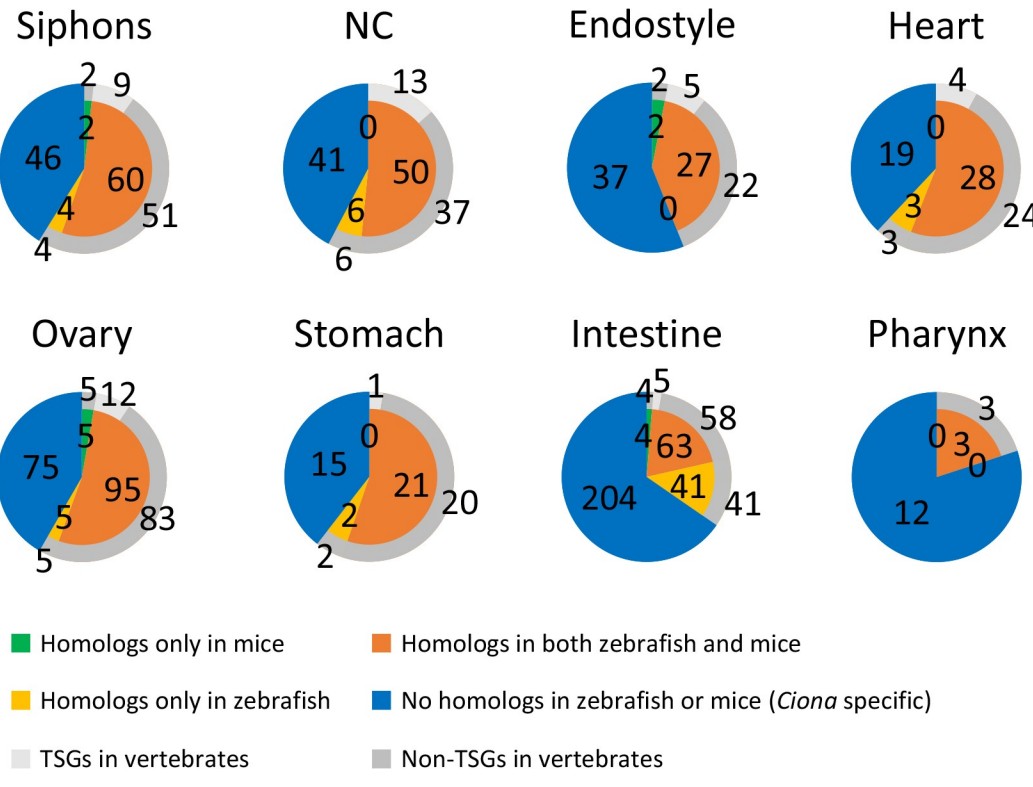

**Fig 2. Number of genes homologous to the *Ciona* TSGs in mice and zebrafish.** Amino acid sequences of the *Ciona* TSGs were blasted against the RefSeq protein database of mice and zebrafish with the e-value set to < 1e-5. The number of genes homologous to mice (green), zebrafish (yellow), or both (orange) in each tissue are shown. The genes without BLAST hits are shown as non-homologous (blue). Among the vertebrate homologs, TSGs and non-TSGs in the vertebrate tissues were also investigated using RNA-seq data of mice (PRJNA267840) and zebrafish (PRJNA255848), and shown in light grey and dark grey, respectively. NC, neural complex.

that expression of the homologous genes in CNS tissues is highly conserved in the evolutionary lineage of chordates. Tissue distributions of the mouse and zebrafish homologs to *Ciona* TSGs were visualized as heat maps and clustered by their expression patterns (Fig 3B). Thirty-two mouse and 27 zebrafish homologs were included in the brain cluster (Fig 3B, orange). Gene ontology (GO) analyses indicated that 18 mouse and 12 zebrafish homologs have characteristic GO terms (biological process) for brain development or function (Fig 3B, *Ciona* IDs in red and S2 Table), and 6 genes of the genes (cholinergic receptors (*Chrnb3* and *Chrm5*), gamma-aminobutyric acid (GABA) receptor (*Gabra6*), genes for synapse organization (*Mdga2* and *Nlgn1*) and brain differentiation (*Otp*)) were predominantly expressed in the brain of both species (Fig 3B, highlighted in yellow and S2 Table). Specific expression of the *Ciona* homologs (KY.Chr10.638, KY.Chr3.84, KY.Chr7.428, KY.Chr9.555, KY.Chr3.577, and KY.Chr14.946) of the 6 genes (*Chrnb3*, *Chrm5*, *Nlgn1*, *Gabra6*, *Mdga2*, and *Otp*) in the neural complex was reproduced by qRT-PCR using 3–4 independent sets of the *Ciona* tissues from that used for RNA-seq (Fig 3C), confirming the reliability of RNA-seq data.

Vertebrate receptors for acetylcholine (*Chrnb3*, *chrnb3a*, *Chrm5*, and *chrm4a*) and GABA (*Gabra6* and *gabra4*) were also specifically expressed in the corresponding brain (Fig 3B), suggesting that fundamental functions of the neurotransmitters acetylcholine and GABA are conserved in chordates. With respect to the other neurotransmitter-related genes, 2 AMPA-type glutamate receptors (KY.Chr2.1128 and KY.Chr3.791) [28], 3 candidate metabotropic-type glutamate receptors (KY.Chr4.1146, KY.Chr12.932, KY.Chr6.541) [29], and 7 monoamine

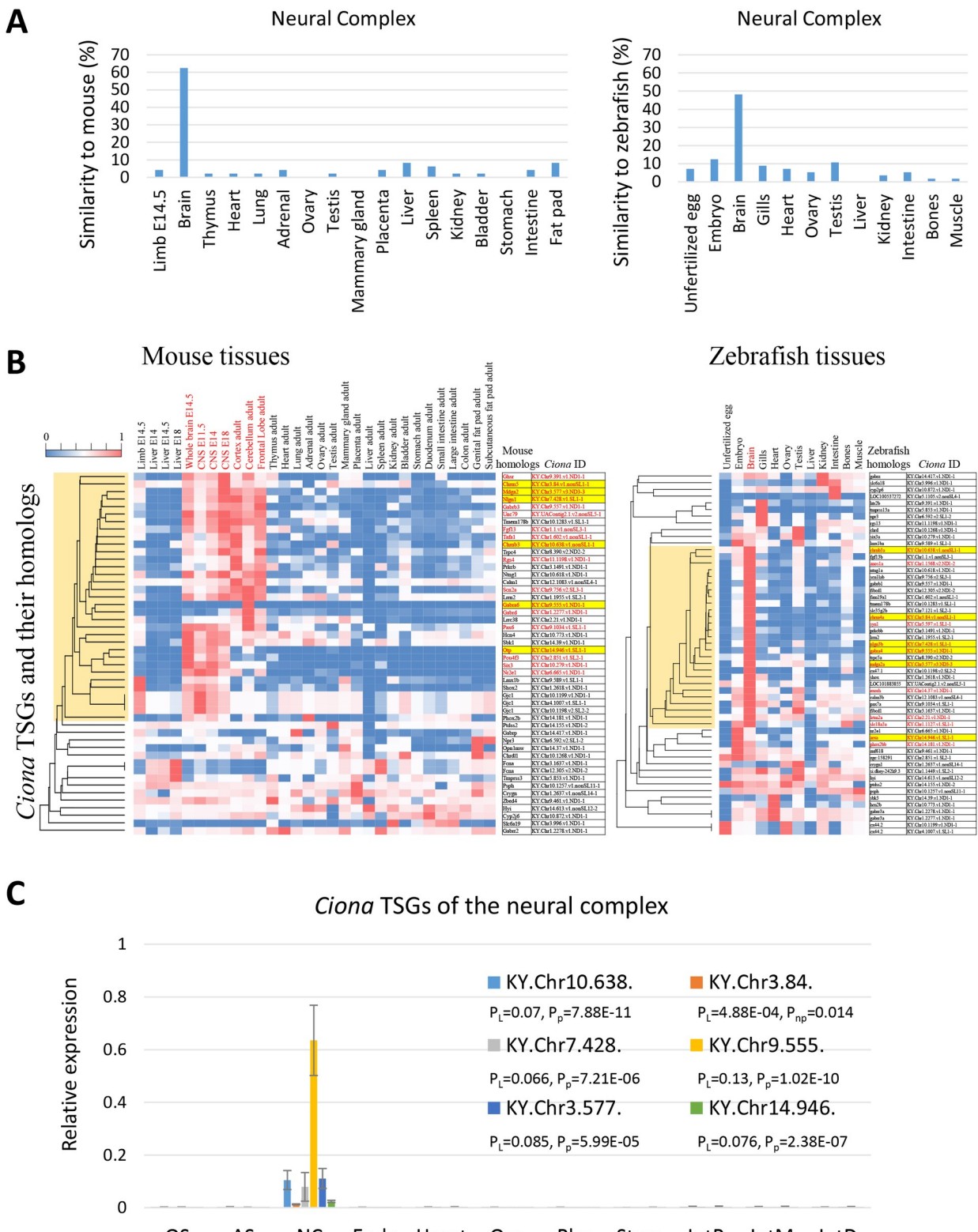

**Fig 3. Comparative analyses of *Ciona* TSGs in the neural complex and their homologs in mice and zebrafish.** (A) Similarities in gene expression patterns between the *Ciona* TSGs in the neural complex and their homologs in mice (left) or zebrafish (right) tissues were calculated. Mouse (62.5%) and zebrafish (48.2%) homologs of *Ciona* neural complex-specific genes showed high expression in the corresponding mouse and zebrafish brains. (B) Clustering of homologous genes in mice and zebrafish by tissue distribution. The heat map shows the expression levels of the

homologs in the 30 mouse tissues (left) and 12 zebrafish tissues (right). The brain clusters are shown in orange. Gene symbols annotated with characteristic GO terms for brain function and development and their *Ciona* homologous IDs are indicated in red. The red IDs in both mice and zebrafish are highlighted in yellow. (C) The neural complex-specific expression of the *Ciona* TSGs was confirmed by qRT-PCR (n = 3–4). Expression is shown relative to the *Ciona* KDEL endoplasmic reticulum protein retention receptor 2 gene (*CiKdelr2*, KY.Chr10.704), which was found to be constitutively expressed among the 9 tissues, according to the RNA-seq analysis. P-values from statistical analyses with the Levene test, Kruskal-Wallis one-way ANOVA, and parametric one-way ANOVA are indicated as $P_L$, $P_{np}$, $P_p$, respectively. AS, atrial siphon; Endo, endostyle; IntD, distal intestine; IntP, proximal intestine; IntM, middle intestine; NC, neural complex; OS, oral siphon; Ova, ovary; Pha, pharynx; Stom, stomach.

receptors (CiHT1-a, Ci5HT1-b, Ci5HT-2, Ci5HT7-a, CiADREβ-a, CiADREβ-b, and CiADREα-2a) [30] were broadly expressed in several tissues (S1 Table). Moreover, most of the neuropeptide genes [14, 15] showed high expression in the neural complex, but were also enriched in the siphons or other tissues (S1 Table). The peptide receptors [31] showed various tissue distributions (S1 Table). These results imply the strict functions of the cholinergic and GABAergic systems in the adult neural complex and broad functions of glutamate, mono-amine, and neuropeptidergic systems in various tissues.

Interactions between immunoglobulin superfamily proteins, Mdga and the synaptic orga-nizing protein Nlgn, are important for regulating the dynamic balance of synapse development in mice [32]. Specific expression of the zebrafish homologs (*mdga2a* and *nlgn3b*) and *Ciona* homologs (KY.Chr3.577 and KY.Chr7.428) suggest some conserved roles in synapse regula-tion. Of note, tissue distributions of the mouse *Otp* homolog and zebrafish *arx* homolog were restricted to the respective mouse and zebrafish brain (Fig 3B). Moreover, mouse homologs of the other transcription factors (*Pax6*, *Pou4f3*, and *Nr2e1*) were also found to be brain- or embryo-specific in the mouse tissues, whereas zebrafish homologs (*pax7a*, *pou4f4*, and *nr2e1*) were not tissue-specific though predominantly expressed in the zebrafish brain or embryo (S1 and S2 Tables). These results suggest that the essential roles of the transcription factors in neu-rodevelopment are conserved in chordates and some of them act as drivers of the brain- or embryo-specific gene expression. In contrast, a mouse homolog (*Six3*) of the other homeobox protein *Ciona* Six3/6 (KY.Chr10.279) was exclusively expressed in the mouse brain; in contrast, the zebrafish homolog (*six3a*) was predominantly expressed in the zebrafish testis (Fig 3B), suggesting divergent roles of Six3a in zebrafish. None of the marker genes for specific neurons in *Ciona* embryos (e.g., *Dmbx* (KY.Chr1.2439) for decussating neurons and *Prop* for Eminens neurons) [7] showed tissue-specific expression in the adult tissues (S1 Table), sug-gesting multifunctionality of these genes or broad distribution throughout the peripheral ner-vous systems. Vertebrate homologs of the chemokine-like protein (*Tafa1* and *tafa1a*) were also found to be brain-specific in both vertebrates (S2 Table), suggesting conserved roles in animal physiology and behavior [33].

Similar to the *Ciona* neural complex, 28.6% of mouse homologs of *Ciona* heart-specific genes were predominantly expressed in the mouse heart, and distinct sets of zebrafish homo-logs were expressed in the zebrafish heart (25.8%) and muscle (25.8%) (S2A Fig). Four verte-brate homologs were found to be TSGs in the vertebrate tissues, and 2 of them (*Adprhl1* and *mybpc3*) were heart-specific (S2 Table). No homologs of heart-specific transcription factors were conserved in mice or zebrafish (S2 Table). Four mouse homologs essential for heart beat-ing and development (*Mybpc3*, *Bmp10*, *Smyd1*, and *Mylk3*) were identified (S2B Fig and S2 Table) and specific expression of the *Ciona* homologs (KY.Chr1.628, KY.Chr14.1196, KY.Chr6.594, and KY.Chr3.1260) was confirmed (S2C Fig). BMP10 has been reported as a ligand for the ALK1 receptor and is important for vasculature development and maintenance in both zebrafish and mice [34]. Additionally, the myosin-interacting protein SMYD1 is essential for sarcomere organization in both species [35]. Mutations in the myosin-binding protein C3

variant (*Mybpc3* and *mybpc3*) and myosin light chain kinase 3 (*Mylk3* and *mylk3*) cause hypertrophic and dilated cardiomyopathy in both species [36–38]. Accordingly, the essential genes for heart development are likely to be conserved in ascidians. Combined with the previous reports showing the transcriptomic similarities among the vertebrate brains and heart tissues [18, 19], the current results suggest that the similarities in gene expression patterns in the brain or heart conform to not only vertebrates but also chordates including ascidians. Consequently, the fundamental functions of neurotransmission and heart beating, and organization of synapses and sarcomeres are likely to be conserved among ascidians, zebrafish, and mice.

## (ii) *Ciona*-unique gene-rich tissues: Ovary, intestine, and pharynx

Surprisingly, expression profiles between *Ciona* TSGs and their homologs in the ovary and intestine were not similar to those in corresponding ovaries and intestines, but rather exhibited similarity to other tissues. Only 4.3% and 14.0% of the mouse and zebrafish homologs of *Ciona* ovary TSGs were highly expressed in the respective ovaries, whereas 30.4% and 14.0% were highly expressed in the mouse and zebrafish brain, respectively (S3A Fig). Similarly, 11.5% and 13.5% of the mouse and zebrafish homologs of *Ciona* intestine TSGs were highly expressed in the respective intestines, while 44.2% and 15.4% were highly expressed in the zebrafish and mouse testis, respectively (S4A Fig). The pattern of TSG expression is likely indicative of the distinct reproductive and nutrient uptake processes among the species. In the ovary, although 12 vertebrate homologs were found to be TSGs in the vertebrate tissues, only 2 of them (*Astl* and *Zar1l*) were ovary specific in the mouse tissues (S2 Table). No homologs of ovary-specific transcription factors were conserved in mice or zebrafish (S2 Table). Moreover, 11 zebrafish homologs were included in the zebrafish ovary-rich cluster that was not found in mice (S3B Fig, pink). Thus, further studies of these genes are expected to be useful for understanding differences in the mechanisms of folliculogenesis and oogenesis between mammals and aquatic animals.

Other clusters of predominant expression with characteristic GO terms were observed in several tissues (S3B Fig, *Ciona* IDs in red). Moreover, 8 genes were found in both mouse and zebrafish, but only one (KY.Chr7.498) shared a similar tissue distribution (high in the brain) (S3B Fig, highlighted in yellow). These results suggest that gene expression for female gametogenesis has diverged in a species-specific fashion. It is noteworthy that the current results are not inconsistent with our previous study demonstrating that the MAP kinase (*CiErk1/2*, KY. Chr6.139), maturation promoting factor (*CiCcnb*, KY.Chr4.1303 and *CiCdk1*, KY.UAContig35), and matrix metalloproteinase (*CiMmp2/9/13*, KY.Chr3.680), which play pivotal roles in the conserved pathway of oocyte maturation and ovulation [26], are multifunctional molecules that are widely distributed in various tissues, and are not specifically expressed in the ovary (S1 Table).

With respect to the intestine, only 5 vertebrate homologs were found to be TSGs in vertebrates (S2 Table). The major cluster in zebrafish included *Ciona*-TSG homologs that were predominantly expressed in the testis, and 5 out of 19 genes harbored characteristic GO terms for meiosis (S4B Fig, pink and S2 Table). A moderate similarity to the mouse testis was also observed (S4A Fig), which may reflect a slight contamination of the invasive or adhesive testis to the *Ciona* intestine. The other 2 clusters of the brain and intestine were also found in both species (S4B Fig, orange), but functional annotations of most zebrafish genes were unavailable. Four common genes (S4B Fig, highlighted in yellow, KY.Chr3.158, KY.Chr8.667, KY. Chr14.625, and KY.Chr4.41) homologous to the mouse homeobox protein *Cdx*, intraflagellar transport protein, *Ttc*, DNA repair protein, *Rad51*, and beta-hexosaminidase, *Hexb*, showed specific expression in the *Ciona* intestine (S4C Fig). However, only the vertebrate *Cdx2* and

*cdx4* genes were specifically expressed in the vertebrate intestine, while others were distributed among various tissues (S4B Fig). No other homologs of intestine-specific transcription factors were found in mice or zebrafish (S2 Table), suggesting that the *Ciona* intestine might have evolved in a *Ciona*-specific manner.

The pharynx is responsible for respiration and food collection, as well as immune responses [11, 39–41]. The current RNA-seq data confirmed the high expression of the immune-complement CiC3 gene (KY.Chr11.1089) and the CiTNFα gene (KY.Chr3.1442) (S1 Table) reported in previous studies [39–41]. These findings are compatible with the view that the pharynx is an immune-responsive organ. However, comparative analysis of the zebrafish and mouse homologs was not performed, given that only 15 TSGs were newly identified, with 12 having no homologous genes in zebrafish and mice (Fig 2). Thus, it is presumed that the pharynx might have evolved in a *Ciona*-specific lineage along with the development of *Ciona*-specific respiration, nutrient uptake, and immune systems.

## (iii) Homologous TSG-rich tissues histologically unrelated to the vertebrate counterparts: Siphons, endostyle, and stomach

Of the TSGs in the siphons, 25.9% and 23.4% of the vertebrate homologs showed high expression in the mouse and zebrafish brain, respectively (S5A Fig). Moreover, characteristic cluster and GO terms for brain function and development were observed (S5B Fig, orange, S2 Table). These results are consistent with previous reports demonstrating that peptidergic neurons are enriched in the siphons and endostyle [12, 13]. Additionally, 24.1%, 23.4%, and 21.9% of the vertebrate homologs were highly expressed in the mouse limb of E14.5 embryo, zebrafish embryo, and zebrafish intestine, respectively (S5 Fig), raising the possibility that siphons retain a group of genes expressed during embryogenesis or have an ancestral function in the intestinal system. Although 6 genes were found as common in mice and zebrafish, only 1 gene (KY.Chr14.962) exhibited orthologous hits from the BLATP analysis of mice and zebrafish (collagen type XII, *Col12a1* and *col12a1a*) and the other 5 genes resulted in different BLAST hits between mice and zebrafish (S5B Fig, highlighted in yellow). Combined with the fact that definite siphon counterparts in the mouse and zebrafish tissues are unclear, these results suggest that siphon-specific genes and their homologs might have evolved in a species-specific lineage with divergent roles in each tissue.

Similar results were observed in the TSGs in endostyle; 28.6% and 18.5% of the vertebrate homologs showed high expression in the mouse and zebrafish brain, respectively (S6A Fig) with a characteristic cluster and GO terms for brain function and development (S6B Fig, orange, S2 Table). One homolog (KY.Chr6.400) of the *Slit* gene, which is important for neural development, was specifically expressed in the endostyle (S6C Fig). The endostyle is believed to share some functions with the vertebrate thyroid gland by the prominent expression of the thyroid-related genes and roles in regulating iodine concentrations [42–45]. The current study confirmed the previous endostyle-specific expression of the *CiVWFL* genes (KY.Chr1.1785 and KY.Chr10.1161) [46] and predominant expression of the thyroid-related transcription factor genes (*Foxe*, KY.Chr5.63 and *Foxq*, KY.Chr3.324) [42] (S1 Table). The *Ciona* galectins (*CiLgals*, KY.Chr4.949 and KY.Chr6.43), the immune-responsive genes expressed in the endostyle [47], were found to be expressed not only in the endostyle but also in the neural complex, pharynx, stomach, intestine and the other tissues (S1 Table). Collectively, the present data support the previous study demonstrating that the endostyle is a thyroid-related organ.

Intriguingly, only 4.8% of the mouse homologs of the *Ciona* stomach-specific genes were highly expressed in the mouse stomach, whereas 28.6% and 33.3% were expressed in the mouse liver and intestine, respectively. Similarly, 34.8%, 21.7%, and 17.4% of the zebrafish homologs

were highly expressed in the zebrafish kidney, intestine, and liver, respectively (Fig 4A). One zebrafish homolog without annotation is the only TSG in vertebrates (S2 Table). The mouse homologs of *Ciona* TSGs in the stomach showed 2 major clusters of highly expressed genes in the intestine and liver with the characteristic GO terms for these tissues (Fig 4B, orange, *Ciona* IDs in red, and S2 Table). Likewise, zebrafish homologs showed 3 clusters in the intestine, liver, and kidney with the characteristic GO terms for these tissues (Fig 4B, orange, *Ciona* IDs in red, and S2 Table). Moreover, 6 homologs (carboxypeptidase A, *Cpa2*, cytochrome P450, *Cyp2*, interferon regulatory factor, *Irf*, and 3 fibrinogen-related genes, *Fgg*, *Fgb*, or *Fcn*a) of the *Ciona* TSGs (KY. Chr8.1333, KY.Chr11.806, KY.Chr14.909, KY.Chr14.910, KY.Chr14.911, and KY.Chr14.393) were found in both mice and zebrafish (Fig 4B, highlighted in yellow) and tissue specificities of the *Ciona* TSGs were confirmed by qRT-PCR (Fig 4C). Therefore, the *Ciona* stomach may play various roles (such as metabolism of protein and low molecular weight compounds, inflammatory responses, and/or blood regulation) similar to the vertebrate intestine, kidney, and liver. It is also noteworthy that the major pancreatic digestive enzymes (alpha-amylase (KY.Chr5.116), lipase (KY.Chr7.356), trypsin (KY.Chr4.1293), chymotrypsin (KY.Chr10.63), and carboxypeptidase (KY.Chr12.113)), which were previously shown to be specifically expressed in the juvenile stomach [48], were highly expressed in the adult stomach as well as in the distal region of the intestine (S1 Table). These findings suggest distinct roles (or substrates) of these enzymes in the distal region of the intestine from the juvenile and/or adult stomach. Combined with the fact that no orthologs for gastric digestive enzymes, such as pepsin or carboxypeptidase E, were found in the *Ciona* genome, the current results support the hypothesis that the *Ciona* stomach may not be a simple structural and functional homolog of the vertebrate stomach [48, 49]. In other words, the *Ciona* stomach might not only function as a "stomach" but also, at least in part, share some features of the pancreas, liver, kidney, and intestine of vertebrates.

## Evolutionary aspects of each tissue in chordates

In this study, tissue similarities based on the comparative analyses of transcriptomic profiles among the species were investigated in stringent criteria for TSGs. Less stringent criteria for TSGs considering the specific expression in two tissues held up similar results in the neural complex, heart, and stomach (S7 Fig). The *Ciona* neural complex and heart were highly similar to the corresponding counterparts in vertebrates (Fig 5A and 5B) in that gene expression patterns for the *Ciona* TSGs and their homologs (e.g., *Chrnb3*, *Otp*, *Mybpc3*, and *Bmp10*, etc.) suggested conserved organization of synapses and sarcomeres, and consequent similarity in their biological roles in neurotransmission and heart beating (Figs 3 and S2). *Ciona* intestine and ovary were not similar to the corresponding vertebrate tissues, but did exhibit similarity to several other vertebrate tissues, implying a divergence of reproductive strategies and/or nutrient uptakes (S3 and S4 Figs). *Ciona* pharynx might have evolved in a *Ciona*-specific lineage, given that most TSGs in the pharynx were *Ciona*-specific (Fig 2). The *Ciona* stomach was more similar to the vertebrate liver, kidney, and intestine rather than the mouse stomach (Fig 5C). Given that *Ciona* homologs of mouse *Cpa2*, *Cyp2*, *Irf*, and fibrinogen-related genes were specifically expressed in the *Ciona* stomach, the *Ciona* stomach might have evolved to play various roles normally attributed to the vertebrate liver, kidney, and intestine, such as metabolism of organic compounds and inflammatory responses (Fig 4).

Of particular interest is that several *Ciona* tissues showed high or moderate similarities to the vertebrate brain in light of TSG expression (Fig 5A). This indicates that some *Ciona* homologs of the highly expressed genes in the vertebrate brain might have diverged as peripheral tissue-specific genes in ascidians. In the case of the *Ciona* ovary, 28 vertebrate homologs of the *Ciona* TSGs were specifically expressed in the mouse and zebrafish brain, but most of these

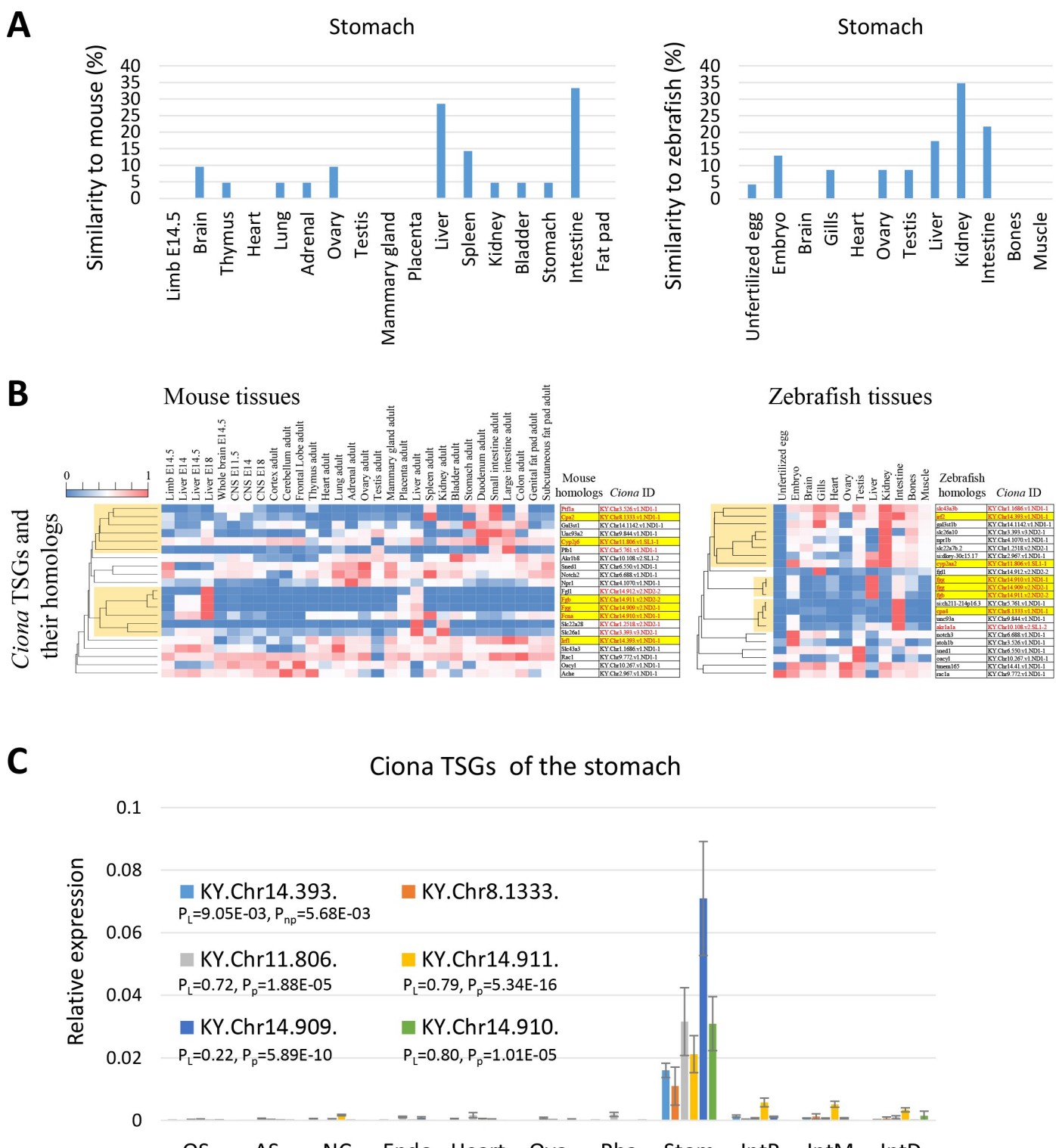

**Fig 4. Comparative analyses of *Ciona* TSGs in the stomach and their homologs in mouse and zebrafish.** (A) Similarities in gene expression patterns between the *Ciona* TSGs in the stomach and their homologs in mouse (left) or zebrafish (right) tissues were calculated as in Fig 3A. Approximately 30% of the homologous genes were highly expressed in the mouse liver and intestine, while 20–35% were highly expressed in the zebrafish liver, kidney, and intestine. (B) Clustering by tissue distribution of the homologous genes in mice and zebrafish. The heat maps are shown as in Fig 3B. Clusters of highly expressed genes in the mouse brain, intestine, zebrafish kidney, liver, and intestine are shown in orange. (C) Stomach-specific expression of *Ciona* TSGs in the stomach was confirmed by qRT-PCR (n = 3–4). Data are presented as in Fig 3C.

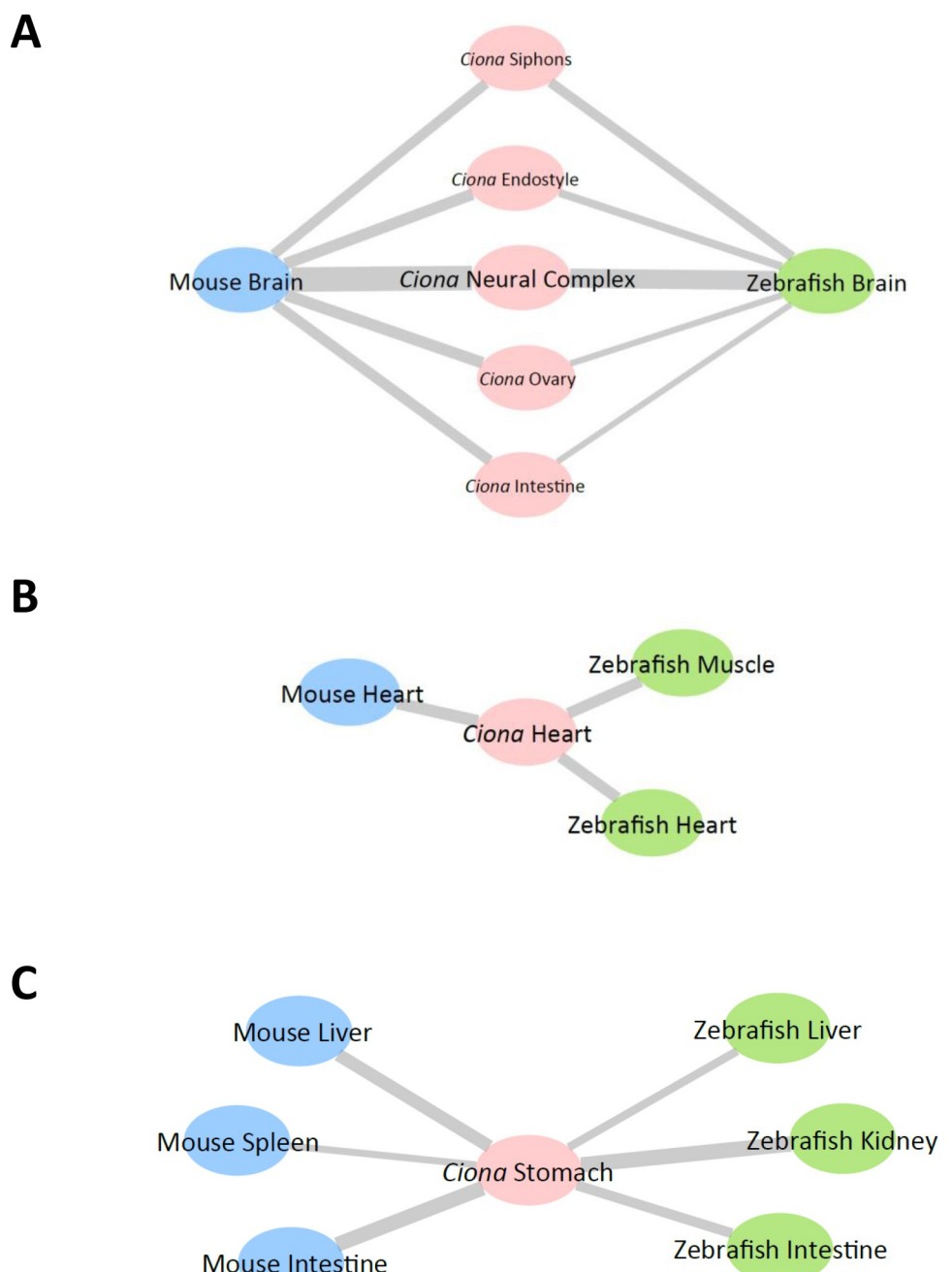

**Fig 5. Schematic summary of the comparative analyses of the brain, heart, and stomach.** Similarities in gene expression patterns between the *Ciona* and zebrafish or mouse tissues illustrated using Cytoscape software (ver. 3.8.2.). *Ciona*, zebrafish, and mouse tissues are shown as pink, green, and blue nodes, respectively. The width of the lines represents the similarity between the tissues. (A) The *Ciona* neural complex and several other tissues showed high similarities to the zebrafish and mouse brain. (B) The *Ciona* heart was similar to the corresponding heart and muscle of vertebrates. (C) The *Ciona* stomach was not similar to the mouse stomach, but rather was similar to several other tissues including the liver, kidney, and intestine.

were not annotated with characteristic GO terms for brain function or development (S2 Table), suggesting that these genes have evolved a functionally distinct lineage. However, 22 vertebrate homologs of the *Ciona* TSGs in the siphons were highly expressed in the vertebrate

brain and 10 of these were annotated with characteristic GO terms for brain function and development (e.g., CNS development [GO:0007417] for *Bcan*, synapse organization [GO:0050808] for *Adgrl2*, and axon extension [GO:0048675] for *sema5a*, etc.) (S2 Table). Such vertebrate homologs harboring GO terms for brain development and function were found in *Ciona* TSGs of the endostyle (e.g., axonogenesis [GO:0007409] for *Slit1*, etc.) and intestine (e.g., dorsal spinal cord development [GO:0021516] for *Uncx*, etc.) (S2 Table). These results suggest that direct or local regulation of peripheral tissues by the peripheral nervous system is more dominant in ascidians than in vertebrates. Such a view is in good agreement with our findings that the *Ciona* siphons and endostyle were similar to the vertebrate brain (S5 and S6 Figs) and also with a previous study revealing that *Ciona* peripheral tissues are regulated by direct projections of the peripheral nervous system [12, 13]. Collectively, while vertebrates might have evolved complicated regulatory systems by the acquisition of a sophisticated brain organ and indirect regulation via the circulatory system of closed vasculature, ascidians might have evolved a simple regulatory system represented by direct or local regulation by the peripheral nervous system or by a simple circulatory system of open vasculature.

In conclusion, we have obtained the transcriptomic profiles and identified TSGs for the adult tissues of an ascidian, *C. intestinalis* Type A (or *C. robusta*), which lies in a critical position on the phylogenetic tree of chordates. We have also evaluated the tissue similarities between ascidians and zebrafish or mice based on the tissue distribution of *Ciona* TSGs and their homologs in vertebrates. The current study provides important insights into the evolutionary lineages of function and development of tissues in chordates, and will pave the way for understanding the conservation and diversification of animal tissues among these species.

## Materials and methods

### RNA extraction, purification, and RNA-seq analyses

Adult ascidians were excised and 9 tissues (oral siphon, atrial siphon, neural complex, endostyle, heart, ovary, pharynx, stomach, and intestine) were collected from more than 4 individuals. The intestine was divided into 3 parts (proximal, middle, and distal). Total RNA was extracted, purified, and treated with DNase as previously described [26]. A total of 500 ng of quality-confirmed RNA was subjected to RNA-seq using a HiSeq1500 (Illumina, San Diego, CA) in rapid mode, as previously described [26]. The resultant reads were aligned to the KY gene model of the *Ciona* cDNA library [3], which was downloaded from the ghost database (http://ghost.zool.kyoto-u.ac.jp/default_ht.html). The expression level for each gene was calculated as gene-specific RPKM. TSGs were determined by exclusive gene expression: RPKM > 1 in a particular tissue and RPKM < 0.5 in all other tissues. TSGs for the siphons and intestine include the genes with RPKM > 1 either in the oral siphon or atrial siphon and in any part of the intestine, respectively. Total reads, mapping rates, accession numbers, and number of TSGs are summarized in Table 1. Raw reads and calculated RPKM for each gene are listed in S1 Table. The raw sequence data have been deposited in the NCBI database (PRJNA731286).

### RNA-seq data analysis

Amino acid sequences for the *Ciona* TSGs were obtained from the ghost database (http://ghost.zool.kyoto-u.ac.jp/default_ht.html). BLASTP was run against the RefSeq protein database of mouse and zebrafish genes, which were downloaded from the NCBI FTP site (https://ftp.ncbi.nlm.nih.gov/). The threshold was set to an e-value of < 1e-5. The resultant BLASTP hits were considered as mouse or zebrafish genes homologous to *Ciona* TSGs, while the *Ciona* genes lacking hits in BLASTP were considered *Ciona* specific. The tissue distribution for the mouse or zebrafish homologs was investigated using public data. RNA-seq data for 30 mouse

tissues (PRJNA267840) and 12 zebrafish tissues (PRJNA255848) were used. Processed data for mouse gene expression with RefSeqID were directly downloaded from NCBI. Zebrafish data were downloaded from the PhyloFish Portal (http://phylofish.sigenae.org/index.html) and the contigIDs were converted to RefSeqID via BLASTN and BioDBnet (https://biodbnet-abcc. ncifcrf.gov/). Expression levels for the mouse or zebrafish homologs of *Ciona* TSGs were normalized from 0 (as in a tissue with the lowest expression level) to 1 (as in a tissue with the highest expression level). The number of highly expressed genes ($> 0.8$) in each tissue was counted and the ratio was indicated as "tissue similarity" between *Ciona* and zebrafish or mice. Gene ontology for highly expressed genes of mouse or zebrafish homologs was investigated in uniprot (https://www.uniprot.org/). Clustering by tissue distribution was performed using R software (ver. 4.0.0, https://www.r-project.org/).

## qRT-PCR

RNA-seq data was confirmed by qRT-PCR using another 3–4 sets of *Ciona* tissues. The qRT-PCR was performed as previously described [26]. In brief, an aliquot of 1 µg of DNase-treated total RNA isolated from *Ciona* tissues was used for the first-strand cDNA synthesis. qRT-PCR was performed using a CFX96 Real-time System and SsoAdvanced™ Universal SYBR Green Supermix (Bio-Rad laboratories, Hercules, CA). The primers are listed in S3 Table. Gene expression levels were normalized to the *Ciona* KDEL endoplasmic reticulum protein retention receptor 2 (*CiKdelr2*, KY.Chr10.704), which was found to be constitutively expressed among 9 tissues according to the RNA-seq analysis.

## Statistical analysis

dCt values for the qRT-PCR of the *Ciona* tissues were used for statistical analyses, as reported elsewhere [50, 51]. The expression level was set to 0 for genes that were not detected, and samples with 2 or more sets below detection were excluded from the statistical analysis. Statistical analyses were performed using R software. We first analyzed using the Levene test and examined the homoscedasticity of each group (tissue). Genes that exhibited equal variances among the tissues were analyzed by a parametric one-way analysis of variance (ANOVA), followed by the Tukey *post hoc* test. Genes that did not show equal variances were analyzed using a non-parametric Kruskal-Wallis one-way ANOVA, followed by the Dunnett test and Bonferroni adjustment. Differences were considered statistically significant at $P < 0.05$. P-values for the Levene test, Kruskal-Wallis one-way ANOVA, and parametric one-way ANOVA are indicated as $P_L$, $P_{np}$. $P_p$, respectively. P-values for the *post hoc* multiple tests are shown in the S1 File.

## Supporting information

**S1 Fig. Validation of the RNA-seq data by referring to previously identified TSGs.** The tissue specificity of the TSGs previously reported by Shoguchi et al., 2011 [27] was confirmed in the current RNA-seq data. AS, atrial siphon; Endo, endostyle; IntD, distal intestine; IntP, proximal intestine; IntM, middle intestine; NC, neural complex; OS, oral siphon; Ova, ovary; Pha, pharynx; Stom, stomach.
(TIF)

**S2 Fig. Comparative analyses of *Ciona* TSGs in the heart and their homologs in mouse and zebrafish.** (A) Similarities between *Ciona* heart and mouse (left) or zebrafish (right) tissues were calculated as in Fig 3A. Approximately 30% and 25% of the homologous genes were highly expressed in the mouse heart and zebrafish heart and muscle, respectively. (B) Clustering by tissue distribution of the homologous genes in mice and zebrafish. The heat maps are

shown as in Fig 3B. (C) The heart-specific expression of *Ciona* TSGs was confirmed by qRT-PCR (n = 3–4). Data are presented as in Fig 3C.
(TIF)

**S3 Fig. Comparative analyses of *Ciona* TSGs in the ovary and their homologs in mouse and zebrafish.** (A) Similarities between *Ciona* ovary and mouse (left) or zebrafish (right) tissues were calculated as in Fig 3A. (B) Clustering by tissue distribution of the homologous genes in mice and zebrafish. The heat maps are shown as in Fig 3B. The zebrafish-ovary cluster is shown in pink.
(TIF)

**S4 Fig. Comparative analyses of *Ciona* TSGs in the intestine and their homologs in mouse and zebrafish.** (A) Similarities between *Ciona* intestine and mouse (left) or zebrafish (right) tissues were calculated as in Fig 3A. (B) Clustering by tissue distribution of the homologous genes in mice and zebrafish. The heat maps are shown as in Fig 3B. The clusters of highly expressed genes in the mouse brain and intestine are shown in orange, and that of the zebrafish testis is shown in pink. (C) The intestine-specific expression of *Ciona* TSGs in the heart were confirmed by qRT-PCR (n = 3–4). Data are presented as in Fig 3C.
(TIF)

**S5 Fig. Comparative analyses of *Ciona* TSGs in the siphons and their homologs in mouse and zebrafish.** (A) Similarities between *Ciona* siphons and mouse (left) or zebrafish (right) tissues were calculated as is Fig 3A. (B) Clustering by tissue distribution of the homologous genes in mice and zebrafish. The heat maps are shown as in Fig 3B. The clusters of highly expressed genes in the vertebrate brain are shown in orange.
(TIF)

**S6 Fig. Comparative analyses of *Ciona* TSGs in the endostyle and their homologs in mouse and zebrafish.** (A) Similarities between *Ciona* endostyle and mouse (left) or zebrafish (right) tissues were calculated as in Fig 3A. (B) Clustering by tissue distribution of the homologous genes in mice and zebrafish. The heat maps are shown as in Fig 3B. The clusters of highly expressed genes in the vertebrate brain are shown in orange. (C) The endostyle-specific expression of *Ciona* TSGs in the endostyle was confirmed by qRT-PCR (n = 3–4). Data are presented as in Fig 3C.
(TIF)

**S7 Fig. Similarities between *Ciona* tissues and mouse or zebrafish tissue in less stringent criteria for TSGs.** (A) In addition to the 56 *Ciona* TSGs in the neural complex, specifically expressed genes in the two tissues including neural complex were considered; 234 *Ciona* genes were screened and the blast hits of 124-vertebrate homologs were analyzed as in Fig 3A. The mouse (38.1%) and zebrafish (34.4%) homologs of *Ciona* neural complex-specific genes showed high expression in the corresponding mouse and zebrafish brains. Similar analyses were performed on the *Ciona* heart (B) and stomach (C). (B) In addition to the 31 TSGs in the *Ciona* heart, 22 vertebrate homologs of the *Ciona* heart- and the other one tissue-specific genes were analyzed. Mouse (25.5%) and zebrafish (25.6% and 18.6%) homologs were highly expressed in the mouse heart and the zebrafish heart and muscle, respectively. (C) In addition to the 23 TSGs in the *Ciona* stomach, 200 vertebrate homologs of the *Ciona* stomach- and the other one tissue-specific genes were analyzed. Mouse homologs were highly expressed in the intestine (23.6%) and liver (21.2%), and zebrafish homologs were expressed in the intestine (23.5%), kidney (21.1%), and testis (21.1%).
(TIF)

**S1 Table. Raw reads and calculated RPKM values for RNA-seq data.**
(XLSX)

**S2 Table. *Ciona* TSGs with GO terms.**
(XLSX)

**S3 Table. Primers used in this study.**
(XLSX)

**S1 File. Summary of statistical analysis results.** Differences were considered statistically significant at P<0.05 (*, P<0.05; **, P<0.01).
(XLSX)

## Acknowledgments

We acknowledge the National Bio-Resource Project for providing ascidians. We are also grateful to Prof. Shigetada Nakanishi for providing fruitful comments regarding the manuscript.

## Author Contributions

**Conceptualization:** Shin Matsubara.

**Data curation:** Shin Matsubara, Tomohiro Osugi, Akira Shiraishi, Azumi Wada.

**Formal analysis:** Tomohiro Osugi, Akira Shiraishi, Azumi Wada.

**Funding acquisition:** Shin Matsubara.

**Investigation:** Shin Matsubara, Tomohiro Osugi, Akira Shiraishi, Azumi Wada.

**Project administration:** Shin Matsubara.

**Software:** Akira Shiraishi.

**Supervision:** Honoo Satake.

**Validation:** Shin Matsubara, Tomohiro Osugi, Akira Shiraishi, Azumi Wada.

**Visualization:** Shin Matsubara, Tomohiro Osugi, Akira Shiraishi, Azumi Wada.

**Writing – original draft:** Shin Matsubara, Honoo Satake.

**Writing – review & editing:** Shin Matsubara, Tomohiro Osugi, Akira Shiraishi, Azumi Wada, Honoo Satake.

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
