## [Decision Letter · Decision Letter 0]

11 Aug 2021

PONE-D-21-19715

Comparative analysis of transcriptomic profiles among ascidians, zebrafish, and mice: insights from tissue-specific gene expression

PLOS ONE

Dear Dr. Matsubara,

Thank you for submitting your manuscript to PLOS ONE. After careful consideration by one reviewer and the Academic Editor, we feel that it has merit but does not fully meet PLOS ONE’s publication criteria as it currently stands. Therefore, we invite you to submit a revised version of the manuscript that addresses the points raised during the review process.

ACADEMIC EDITOR comments:

This is systematic study comparing the gene expression profiles in adult Ciona robusta organs to those in the organs of zebra fish and mice.  They authors obtained a new adult Ciona organ RNAseq dataset and used clear and strict criteria to first identify tissue specific genes (TSGs) in Ciona.  They then proceed to investigate the expression levels of the orthologs of these TSGs in mice and zebrafish organs based on existing RNAseq datasets.  Striking similarities of expression profiles were observed for brain and heart, while similarities for other organs were less apparent or almost non-existing.  Based on this the authors concluded that the molecular machineries of some organs are highly conserved (brain and heart), while others are either species specific (e.g. ovary) or spread out over several organs (Ciona stomach shows similarty to mouse/zebra fish liver and other metabolic organs).  Overall, this study is a useful resource for scientists with an interest in the evolution of organ systems, in particular at the transition from invertebrates to vertebrates. 

There are three minor concerns that should be addressed by making respective changes to the manuscript: 

What is missing in this manuscript is some information whether (and how many) of the orthologs of Ciona TSGs would also fulfil the very same TSG criterion in one or both of the two vertebrates – this could be easily tested and would strengthen the central role of individual genes in these tissues. It would be interesting to see whether any of the TSGs is a transcription factor as those would be the drivers of tissue specific gene expression.  Thus I recommend to look for this among the TSGs and the expression of the orthologs in mice and zebra fish. The criteria for TSGs are quite stringent (which is good for clarity), but I was wondering what happens when genes that are highly specifically in two (and not just one) Ciona tissue are considered.  Do they findings hold true?  No extensive test need to be done on this but some discussion about how the similarities between tissue hold up when less stringent criteria are applied would be of interest for the readers. 
Indicate which changes you require for acceptance versus which changes you recommendAddress any conflicts between the reviews so that it's clear which advice the authors should followProvide specific feedback from your evaluation of the manuscript

We look forward to receiving your revised manuscript.

Kind regards,

Sebastian D. Fugmann, Ph.D.

Academic Editor

PLOS ONE

Journal Requirements:

"We acknowledge the National Bio-Resource Project for providing ascidians. We are also grateful to Prof. Shigetada Nakanishi for providing fruitful comments regarding the manuscript. This work was supported in part by grants from the Japan Society for the Promotion of Science to SM (JP19K16182)."

"SM, JP19K16182, Japan Society for the Promotion of Science (https://www.jsps.go.jp/english/index.html)

The funders had no role in study design, data collection and analysis, decision to publish, or preparation of the manuscript"

Reviewers' comments:

Reviewer's Responses to Questions

**Comments to the Author**

1. Is the manuscript technically sound, and do the data support the conclusions?

Reviewer #1: Yes

2. Has the statistical analysis been performed appropriately and rigorously? 

Reviewer #1: Yes

3. Have the authors made all data underlying the findings in their manuscript fully available?

Reviewer #1: Yes

4. Is the manuscript presented in an intelligible fashion and written in standard English?

Reviewer #1: Yes

5. Review Comments to the Author

Reviewer #1: This is an interesting report and a good fit for PLoS ONE. I do not see any major issues with the data collection or analysis, however I am not a hands-on expert on the statistics used, thus I would defer to other reviewers opinions.

6. PLOS authors have the option to publish the peer review history of their article (what does this mean?). If published, this will include your full peer review and any attached files.

Reviewer #1: No

---

## [Author Response · Author response to Decision Letter 0]

10 Sep 2021

Response to the academic editor

ACADEMIC EDITOR comments:

This is systematic study comparing the gene expression profiles in adult Ciona robusta organs to those in the organs of zebra fish and mice. They authors obtained a new adult Ciona organ RNAseq dataset and used clear and strict criteria to first identify tissue specific genes (TSGs) in Ciona. They then proceed to investigate the expression levels of the orthologs of these TSGs in mice and zebrafish organs based on existing RNAseq datasets. Striking similarities of expression profiles were observed for brain and heart, while similarities for other organs were less apparent or almost non-existing. Based on this the authors concluded that the molecular machineries of some organs are highly conserved (brain and heart), while others are either species specific (e.g. ovary) or spread out over several organs (Ciona stomach shows similarty to mouse/zebra fish liver and other metabolic organs). Overall, this study is a useful resource for scientists with an interest in the evolution of organ systems, in particular at the transition from invertebrates to vertebrates. 

There are three minor concerns that should be addressed by making respective changes to the manuscript: 

1.　What is missing in this manuscript is some information whether (and how many) of the orthologs of Ciona TSGs would also fulfil the very same TSG criterion in one or both of the two vertebrates – this could be easily tested and would strengthen the central role of individual genes in these tissues. 

<Our response>

Thank you for your constructive comment. We have investigated the tissue specificity of the vertebrate homologs of Ciona TSGs and found that some vertebrate homologs were also tissue-specific. According to the results, we have revised Fig. 2, S2 Table, and the corresponding parts of the main text (lines 161-171, 233-240, 247-249, 252-254, 281-283, 300-301, and 359-360) and figure legend (lines 154-156) in the revised manuscript.

2.　It would be interesting to see whether any of the TSGs is a transcription factor as those would be the drivers of tissue specific gene expression. Thus I recommend to look for this among the TSGs and the expression of the orthologs in mice and zebra fish. 

<Our response>

We have searched the tissue-specific transcription factors in the Ciona tissues and investigated the tissue distribution of their vertebrate homologs. We have found that several vertebrate homologs of the neural complex-specific transcription factors (Otp, arx, Pax6, pax7a, Pou4f3, pou4f4, Nr2e1, and nr2e1) were also specific or predominant in the vertebrate brain or embryo. On the other hand, in the other tissues, none or quite a small number of tissue-specific transcription factors and their vertebrate homologs exhibited conserved tissue distribution between Ciona and vertebrates. These results suggest that some tissue-specific transcription factors act as drivers of tissue-specific gene expression only in the neural complex (brain) but not in the other tissues. We have included these data in the “TFs” tab of the S2 Table and explained them in the revised manuscript (lines 233-240, 254-255, 283-284, and 312-313).

3.　The criteria for TSGs are quite stringent (which is good for clarity), but I was wondering what happens when genes that are highly specifically in two (and not just one) Ciona tissue are considered. Do they findings hold true? No extensive test need to be done on this but some discussion about how the similarities between tissue hold up when less stringent criteria are applied would be of interest for the readers.

<Our response>

Thank you for your constructive comment. We have investigated Ciona TSGs in less stringent criteria considering specific expression in two tissues and obtained similar results, although slight spreading out of the tissue similarities were observed. We have added the data as S7 Fig and described it in the revised manuscript (lines 397-400 and 742-755).

---

## [Editor Report · Decision Letter 1]

13 Sep 2021

Comparative analysis of transcriptomic profiles among ascidians, zebrafish, and mice: insights from tissue-specific gene expression

PONE-D-21-19715R1

Dear Dr. Matsubara,

We’re pleased to inform you that your manuscript has been judged scientifically suitable for publication and will be formally accepted for publication once it meets all outstanding technical requirements.

Kind regards,

Sebastian D. Fugmann, Ph.D.

Academic Editor

PLOS ONE
---

## [Editor Report · Acceptance letter]

16 Sep 2021

PONE-D-21-19715R1 

Comparative analysis of transcriptomic profiles among ascidians, zebrafish, and mice: insights from tissue-specific gene expression 

Dear Dr. Matsubara:

I'm pleased to inform you that your manuscript has been deemed suitable for publication in PLOS ONE. Congratulations! Your manuscript is now with our production department. 

Kind regards, 

on behalf of

Dr. Sebastian D. Fugmann 

Academic Editor

PLOS ONE